# User-Centered Evaluation of the Wearable Walker Lower Limb Exoskeleton; Preliminary Assessment Based on the Experience Protocol

**DOI:** 10.3390/s24165358

**Published:** 2024-08-19

**Authors:** Cristian Camardella, Vittorio Lippi, Francesco Porcini, Giulia Bassani, Lucia Lencioni, Christoph Mauer, Christian Haverkamp, Carlo Alberto Avizzano, Antonio Frisoli, Alessandro Filippeschi

**Affiliations:** 1Institute of Mechanical Intelligence and Department of Excellence in Robotics & AI, Scuola Superiore Sant’Anna, 56127 Pisa, Italy; f.porcini@santannapisa.it (F.P.); g.bassani@santannapisa.it (G.B.); c.avizzano@santannapisa.it (C.A.A.); a.frisoli@santannapisa.it (A.F.); a.filippeschi@santannapisa.it (A.F.); 2Institute of Digitalization in Medicine, Faculty of Medicine and Medical Center—University of Freiburg, 79106 Freiburg, Germany; vittorio.lippi@uniklinik-freiburg.de (V.L.); christian.haverkamp@unikilinik-freiburg.de (C.H.); 3Clinic of Neurology and Neurophysiology, Medical Centre—University of Freiburg, Faculty of Medicine, University of Freiburg, Breisacher Straße 64, 79106 Freiburg im Breisgau, Germany; 4Wearable Robotics S.r.L., 56010 Pisa, Italy; l.lencioni@wearable-robotics.com

**Keywords:** wearable robotics, human factors and human-in-the-loop, human performance augmentation, wearable sensor networks, physical human–robot interaction

## Abstract

Using lower limb exoskeletons provides potential advantages in terms of productivity and safety associated with reduced stress. However, complex issues in human–robot interactions are still open, such as the physiological effects of exoskeletons and the impact on the user’s subjective experience. In this work, an innovative exoskeleton, the *Wearable Walker*, is assessed using the EXPERIENCE benchmarking protocol from the EUROBENCH project. The *Wearable Walker* is a lower-limb exoskeleton that enhances human abilities, such as carrying loads. The device uses a unique control approach called *Blend Control* that provides smooth assistance torques. It operates two models simultaneously, one in the case in which the left foot is grounded and another for the grounded right foot. These models generate assistive torques combined to provide continuous and smooth overall assistance, preventing any abrupt changes in torque due to model switching. The EXPERIENCE protocol consists of walking on flat ground while gathering physiological signals, such as heart rate, its variability, respiration rate, and galvanic skin response, and completing a questionnaire. The test was performed with five healthy subjects. The scope of the present study is twofold: to evaluate the specific exoskeleton and its current control system to gain insight into possible improvements and to present a case study for a formal and replicable benchmarking of wearable robots.

## 1. Introduction

In recent years, lower limb exoskeletons (LLEs) have emerged as groundbreaking innovations, impacting society as both medical devices and assisting devices in industry and logistics.

In the medical field, these wearable robotic devices are designed to support the lower limbs, enabling users to stand, walk, and even regain a semblance of natural movement. The usefulness and adoption of lower limb exoskeletons have the potential to reshape the lives of millions, transcending physical limitations and ushering in a new era of mobility.

In the fast-paced world of industry and logistics, the adoption of lower limb exoskeletons is emerging as a transformative force, revolutionizing the way tasks are performed and redefining the capabilities of human workers. These wearable robotic devices, originally designed to aid individuals with mobility challenges, have found a unique niche in the industrial and logistics sectors, in which they have the potential to enhance productivity, reduce fatigue, and minimize the risk of work-related injuries. By supporting and redistributing the load and reducing the strain on the lower limbs and back, these devices can contribute to a significant decrease in musculoskeletal injuries, fostering a safer and healthier working environment.

The adoption of LLEs in both fields faces barriers related to the ergonomics of the devices, their ability to support productivity, and their cost [1,2,3]. The trade-off between LLE performance and cost has led to design simplifications, such as the reduction in degrees of freedom (DoFs) and actuated DoFs, which may impact the usability of the devices and limit the adoption of LLEs [4]. Therefore, the effects of such simplifications need a thorough evaluation to assess the benefits for the user as a first step towards adoption. In recent years, there have been many attempts to evaluate LLE, especially based on kinetic and physiological measures gathered both in the laboratory and in field experiments [5,6].

The EUROBENCH project has made a significant step towards standardization of exoskeleton assessment [7]. This project fostered the development of a benchmarking facility where LLEs can be evaluated using custom or preliminary validated experimental protocols. In fact, this project funded both the development of testing protocols and the evaluation of existing exoskeletons, thus allowing for a fair comparison of different platforms, which is of great value for the scientific community.

In the EXOSMOOTH project [8,9], we targeted the evaluation of the *Wearable Walker* LLE when using an innovative control strategy [10] for assisted walking while carrying loads. To this end, we opted for a psychophysiological assessment of our exoskeleton using the EXPERIENCE protocol [11,12,13]. This protocol is user-centered, as it ties the evaluation of the LLE to the user’s reactions to its use both in terms of physiological measures such as stress, fatigue, and attention and through a physiological assessment based on a questionnaire on functionality, acceptability, perceptibility, and usability.

This paper reports the preliminary results of this evaluation that show how the exoskeleton does not significantly increase stress on the user in assisted walking, that it is positively evaluated by users. At the same time, there is room for improvement in terms of changes in physiological parameters.

## 2. Materials and Methods

### 2.1. The Wearable Walker Exoskeleton

The *Wearable Walker*, depicted in Figure 1, stands as an innovative, powered lower limb exoskeleton designed to offer assistance to post-stroke injured patients in their daily activities while simultaneously enhancing human capabilities, particularly in load-carrying tasks. The exoskeleton has a rigid structure consisting of seven interconnected links, each featuring two active degrees of freedom (DoFs) per leg—specifically, hip and knee flexion/extension—and two non-actuated DoFs per leg—ankle flexion/extension and ankle inversion/eversion, with the latter remaining unsensorized.

The human–exoskeleton interfaces are placed at the pelvis/shoulders, the thighs, and the feet. The first allows the user to sustain the weight of the exoskeleton when assistance is not activated and constrains the back to follow the motion of the user’s pelvis. Interfaces at the thighs have two DoFs, one translational along the thigh link of the exoskeleton and a rotational one whose axis is aligned to the flexion axes of the exoskeleton (see Figure 1). This solution allows differently-sized users to wear and comfortably use the *Wearable Walker*. Finally, users can place their own shoes on the exoskeleton’s feet and tie them to the exoskeleton’s feet by means of belts. Therefore, users can raise their heels during the stance phase of gait, making walking more comfortable and avoiding the user’s perception that their feet are glued to the ground when they need to quit the stance phase.

Standing at a height of 110 cm, a width of 65 cm, and weighing 15 kg (excluding batteries), the *Wearable Walker*’s physical dimensions make it a versatile and robust device for diverse applications. Table 1 provides comprehensive details on link lengths and masses, contributing to a comprehensive understanding of the exoskeleton’s physical attributes.

Utilizing brushless torque motors, specifically, the RoboDrive Servo ILM 70 × 18 Robodrive GMBH, Seefeld, Germany, coupled with a tendon-driven transmission system comprising a screw and a pulley, each actuated joint experiences a remarkable reduction ratio of 1:73.3 and ensures high efficiency exceeding 90%. This direct motion performance makes efficiency in reverse motion high enough to guarantee good back driveability. The sensory capabilities of the *Wearable Walker* include Hall-effect sensors on each joint, along with encoders on every motor shaft. In addition, the four force sensor resistors in each shoe insole provide valuable feedback during weight-bearing activities. Moreover, four one-axis load cells are suitably positioned on the thigh interface of each leg, measuring interaction forces in the sagittal plane and perpendicular to the thighs. These forces represent the primary interactions between the exoskeleton and the user during straight walking, ensuring optimal support and adaptability. Signal acquisition and motor control are realized by two boards placed on each thigh, as shown in Figure 2. The exoskeleton’s backlink serves as the hub for essential components. Housing the power electronics group, it accommodates a 48 V, 14 Ah battery weighing 4.3 kg and a voltage conversion board. Meanwhile, the control unit comprises a computer operating Simulink Real-Time running at 5 KHz frequency and a Wi-Fi communication module. This sophisticated control unit establishes communication with the low-layer control module of each leg through the EtherCAT protocol, facilitating seamless coordination and synchronization. Figure 2 shows the communication blocks along with the computing units. Additionally, a host PC can remotely control the Simulink Real-Time PC through the Wi-Fi module, allowing for real-time adjustments and monitoring.

### 2.2. Control Strategy

There are three layers in the control strategy [10]. Based on two look-up tables—one based on joint velocities and the other on joint positions—the lowest layer compensates for friction and ripples. The middle layer makes the assumption that the exoskeleton can only be in one of two single-stance configurations: either right-foot or left-foot-grounded, or in a double-stance that combines the two single-stance configurations. Therefore, two kinematic models and related compensation models are applied for the two single-stance phases. Equation (Equation 1) provides the joint angle vectors for both the left-foot-grounded stance and the right-foot-grounded single stance, with reference to Figure 1:(1)$$\begin{array}{cc} q & ={\left[q_{RH}\;q_{RK}\;q_{RA}\;q_{LH}\;q_{LK}\;q_{LA}\right]}^{T}\\ q_{L} & ={\left[q_{LA}\;q_{LK}\;q_{LH}\;q_{RH}\;q_{RK}\right]}^{T}\\ q_{R} & ={\left[q_{RA}\;q_{RK}\;q_{RH}\;q_{LH}\;q_{LK}\right]}^{T}, \end{array}$$
where q, $q_{L}$, and $q_{R}$ are vectors of joint variables suitably arranged to compute inertial, gravity, friction, and ripple compensations. This middle layer provides inertia and gravity compensations based on these kinematic models. Equation (Equation 2) reports the overall compensation for each single-stance model, which includes low-level control:(2)$$\begin{array}{cc} \tau_{L} & =B_{L}\left(q_{L}\right){\ddot{q}}_{L}+G_{L}\left(q_{L}\right)+\tau_{F}\left(\dot{q}\right)+\tau_{r}\left(q\right)\\ \tau_{R} & =B_{R}\left(q_{R}\right){\ddot{q}}_{R}+G_{R}\left(q_{R}\right)+\tau_{F}\left(\dot{q}\right)+\tau_{r}\left(q\right), \end{array}$$
where ${\ddot{q}}_{L}$ and ${\ddot{q}}_{R}$ are the joint acceleration vectors, $B_{L}$ and $B_{R}$ are the inertia matrices, $G_{L}$ and $G_{R}$ are the gravity vectors, and $\tau_{F}$ and $\tau_{r}$ are, respectively, the friction and the ripple compensation. Single-stance assistance, thanks to gait segmentation, is combined by means of high-level control (see Equation (Equation 2)) to build the final assistive torques τ provided to the user. Figure 2 shows the blocks that provide the ripple, friction, inertia, and gravity components of the assistive torques highlighted in red.

#### 2.2.1. Gait Segmentation

The suggested method takes advantage of a classifier that produces a continuous gait phase class that ranges from +1 (left-footed single stance) to −1 (right-footed single stance) based on a linear regression of the joint angles. Every new user requires training of the regressor, which depends on the user. As a result, a quick ($≃2^{\prime }$) training phase has been added, during which the user must perform the following tasks: (1) three full swings with the left leg (left-foot-grounded single stance); (2) three full swings with the right leg; and (3) walking on a treadmill at a variable speed of between one to three kilometers per hour using 0.5 km/h speed steps. In this phase, gait stages are labeled using sole pressures. With the help of labeled data, the linear regressor is trained to identify the subject gait phases by reducing the classification error. Therefore, the regression problem is framed as follows given the labeled data $p\left(t\right)\in R^{T}$ (where *T* is the number of samples in the training set) and the joint angles over time $Q\left(t\right)\in R^{T\times n}$ (where n=6 is the number of joints):(3)$$Y=\underset{Y\in R^{n}}{argmin}{\left\|YQ\left(t\right)-p\left(t\right)\right\|}_{2}.$$

#### 2.2.2. Blend Control

To ensure a seamless aid transition between the gait phases, $\tau_{R}$ and $\tau_{L}$ can be blended based on the regression’s result *Y*. The suggested method blends the compensatory torques using two continuously varying gains between 0 and 1. Therefore, these two gains are defined as follows:(4)$$\begin{array}{cc} \gamma_{L}\left(q\right) & =\frac{1}{2}\left(Yq+1\right)\in \left[0,1\right]\subset R\\ \gamma_{R}\left(q\right) & =1-\gamma_{L}\left(q\right), \end{array}$$
and the overall assistance is given as in Equation (Equation 5):(5)$$\begin{array}{cc} \tau & =\gamma_{L}\left(q\right)\tau_{L}+\gamma_{R}\left(q\right)\tau_{R}\\ \Rightarrow \tau & =\gamma_{L}\left(q\right)\left(B_{L}\left(q_{L}\right){\ddot{q}}_{L}+G_{L}\left(q_{L}\right)\right)\\ & +\gamma_{R}\left(q\right)\left(B_{R}\left(q_{R}\right){\ddot{q}}_{R}+G_{R}\left(q_{R}\right)\right)+\tau_{F}\left(\dot{q}\right)+\tau_{r}\left(q\right). \end{array}$$

The total assistance is continuous since the modulation coefficients $\gamma_{L}$ and $\gamma_{R}$ are continuous. This ensures seamless support regardless of the gait phase shift. Unlike smoothed finite state machine-based controls, this method ensures that the assistance is always consistent with the gait phase, even when changes occur. It is anticipated that this will increase transparency and reduce interaction pressures. *Blend Control* is the term used to describe this support technique [10].

### 2.3. Experimental Protocol

#### 2.3.1. The EXPERIENCE Sub-Project

The EXPERIENCE sub-project aims to provide developers of lower limb exoskeletons and clinical experimenters with a benchmarking method capable of catching the USP of end users by using a newly developed multi-factor questionnaire called the EXPERIENCE Questionnaire (EQ), as well as algorithms to extract psychophysiological indicators based on physiological data. The main outcomes of the sub-project include benchmarking software for the user-centered assessment of exoskeleton-assisted overground and treadmill-based walking, divided into two modules:Module 1: Based on the newly developed multi-factor questionnaire;Module 2: Algorithms to extract psychophysiological indicators starting from the physiological measures gathered as described in the following paragraph.

#### 2.3.2. Physiological Measures

The EXPERIENCE protocol requires the sensors to be used to gather physiological measures [13]. Those included in the protocol are the galvanic skin response (GSR), heart rate (HR), heart rate variability (HRV), and respiration rate (RR), which are the physiological performance indicators (PIs). These were monitored using two commercially available devices: the ZephyrBioModule 3 sensor from Medtronic (Minneapolis, MN, USA), which employs a chest belt to measure RR and electrocardiogram (ECG), and the Shimmer sensor, which captures GSR via electrodes positioned on the index and middle fingers. The ZephyrBioModule 3 sensor provided insights into respiratory patterns and cardiac activity, while the Shimmer sensor allowed for precise monitoring of GSR, reflecting sympathetic nervous system activity.

#### 2.3.3. Participants

Five subjects have been enrolled in this study (4 males, 1 female, aged 32.2±6.98 years, 180.4±7.77 cm height), attending the experiments at the exoskeleton benchmarking facility of Los Madronos Hospital in Brunete, Madrid, supported by the EUROBENCH European project. Before starting the experiment, all subjects signed the informed consent given by the experimenter. This study has been approved by the Scuola Superiore Sant’Anna and the CSIC ethical committees.

#### 2.3.4. Protocol

The EXPERIENCE experimental protocol for each assessment session is based on the step reported below. The experimenter has to:Place the two physiological sensors onto the subject’s body:
(a)ZephyrBioModule 3 sensor, connected to a Zephyr™ band featuring ECG and breathing sensors, aligning conductive ECG sensors with the center of the chest and the breathing sensor with the left side of the thorax, slightly moistening the pad surfaces with water to promote conductivity;(b)Shimmer GSR sensor (Dublin, Ireland), attached to the patient’s wrist using the adjustable strap. The electrodes must be placed on the back of the index and middle finger of the non-dominant hand.Start data collection software and start recording of physiological data.Ask the subject to sit with eyes closed and let her/him relax.Mark the beginning of the recording of the seated baseline on the data collection software and mark the stop when finished. Time duration of recording is 4 min and has to be measured by a chronometer.Place the exoskeleton onto the subject’s body and let it relax.Flag the start of the recording of the standing baseline on the data collection software and flag the stop when finished.Start the robot-assisted walking session and wait until a steady-state condition is reached. The steady-state condition is reached when assistance parameters are not changed anymore and the subject is walking comfortably without any major change.Flag the start of the recording of the walking condition on the data collection software and flag the stop when finished. Time duration of recording is 16 min and has to be measured by a chronometer.Stop the walking session and stop data collection software.Remove physiological sensors and the exoskeleton.Start the questionnaire compilation (see Section 2.4).

### 2.4. Questionnaire

The EUROBENCH benchmarking team created the multifactorial EQ to evaluate the lower limb exoskeleton in situations involving overground and treadmill rehabilitation. Thanks to psychological theories on technology acceptance and usability quality models in the literature, factors that are appropriate for evaluating user experience when using assistive lower limb exoskeletons and rehabilitation during usage have been identified. Four factors (usability, acceptability, perceptibility, and functionality) comprise the 16 sub-factors that we found (Table 1). There are 132 items in total on the questionnaire. A seven-point Likert-type scale, ranging from 1 for “I strongly disagree” to 7 for “I strongly agree”, must be used to assess each question. A new reversed score of 7+1−5=3 is mapped for the reversed item “The exoskeleton is useless”, which has a score of 5. Other items have their scores adjusted as well. In order to assess the user’s consistency in assigning scores to items that are highly similar, the EQ has a control sub-scale called the consistency scale. Control items—rephrased versions of previously administered items—are concealed inside the EQ in order to achieve this goal. Similar item scores can be compared, and differences between the scores may suggest that the subjects’ responses are not entirely reliable. The EUROBENCH benchmarking team provided the EXPERIENCE questionnaire software, which was used to gather data.

#### Questionnaire PIs

The adopted questionnaire allowed us to obtain 4 PIs:Usability, which is defined as how effectively, efficiently, and satisfactorily users can use the exoskeleton to achieve specific goals.Acceptability, which refers to the users’ perception of the exoskeleton and their willingness to incorporate it into daily life.Perceptibility, which measures the emotional and perceptual impact of using the exoskeleton and its effect on quality of life. A high value indicates a positive influence.Functionality, which assesses how the user perceives the exoskeleton in terms of ease of learning, flexibility of interaction, reliability, and workload.

The calculation of PIs is based on three-level scoring ([12,13]):(6)$$\begin{array}{cc} ss & =\frac{1}{N_{i}}\sum_{k=1}^{N_{i}}is_{k}\;fs=\frac{2}{N_{s}\left(N_{s}-1\right)}\sum_{k=1}^{N_{s}}w_{k}ss_{k}, \end{array}$$
with $N_{i}$ being the total number of items belonging to each sub-factor, $N_{s}$ the total number of sub-factors belonging to each factor, and $N_{f}$ the total number of factors. The weights $w_{k}$ are derived from the comparison between pairs of sub-factors (pairwise comparison) administered to the subject under testing. Sub-factors are sorted based on the number of collected preferences so that the weights are consequently assigned. The fs formula is used to calculate PIs.

The computation of the user’s consistency is based on the difference between the scores assigned to two similar items (the original item included in the EQ and the control rephrased item hidden within it). This difference will be indicated as control item discrepancy. A total of 16 control items has been included. An item and its control are considered to be scored in a consistent way if the control item discrepancy is in the range [−1,1]. A higher control item discrepancy causes a proportional decrease in the total consistency score (maximum 100%).

### 2.5. Psychophysiological PIs

The following are the four psychophysiological PIs:Stress is characterized as a condition of mental or emotional tension brought on by unfavorable events. A high PI score suggests that using a robot can be stressful.Energy means the quantity of energy required to perform bodily tasks. A high PI number suggests that using a robot involves a lot of work.Attention describes the level of conscious and ongoing user involvement in the work. A high value for this PI suggests that using a robot demands careful attention.Fatigue is distress typically caused by the muscles in the body becoming too tired to do a task. A high PI score suggests that the robot use induces fatigue.

The ECG data were processed to derive the power distribution in the low frequency (LF) band (0.04–0.15 Hz). The GSR signal was split into two parts: the skin conductance response (SCR), which is an event-dependent, phasic, and highly responsive parameter, and the skin conductance level (SCL), which is the tonic level in the absence of any specific environmental event. For the purpose of extracting the four PI indicators of stress, fatigue, energy expenditure, and attention, a fuzzy logic approach method was used ([12,13]). Only the last 5 min of the recording were taken into account, and the physiological signals from the walking phase were normalized in relation to the information gathered during the sitting phase. The six inputs (HR, RMSSD, RR, SCR, SCL, and LF) and four outputs (PIs) were used in the fuzzy logic technique. The membership functions were created for each physiological signal by accounting for the total number of occurrences in all of the data that were gathered. Specifically, three levels (low, medium, and high) were introduced for the model’s inputs and outputs. The IF/THEN rules were developed using physiological signal variation trends that were discovered through a thorough review of the scientific literature. In order to estimate the four PIs, these algorithms integrate all of the physiological data that were provided.

## 3. Results

The analysis of the proposed PIs resulted in three main results concerning the analysis of the physiological signals as reported in Figure 3, the psychophysiological metrics in Figure 4, and the questionnaire answers plotted in Figure 5. The physiological measures showed no differences between the SIT and SIT EXO phases for HR, RR, and HRV, while it showed a reduction in the GSR measure from 630 ms to 185 ms. Instead, between the SIT EXO and WALK phases, an increase in HR (from 80 bpm to 118 bpm) and RR (from 19 bpm to 31 bpm) occurred, while GSR and HRV did not show any significant variation.

Regarding the psychophysiological PIs, levels of stress and attention do not vary during the execution of the task (equal to, respectively, 0.48 and 0.47), while fatigue and energy increase halfway through the experiment and drop in the end. As a last result, the exoskeleton received moderately positive feedback for all factors, with the best scores in acceptability (mean μ=5.19 and STD σ=1.31) and perceptibility (μ=5.25, σ=1.07). Usability seems to be the most critical aspect, ranking slightly above the average (μ=4.59, σ=1.06) while functionality lies halfway between them (μ=4.94, σ=0.99).

## 4. Discussion and Conclusions

The benchmarking platform allowed us to assess the exoskeleton’s performance from different points of view. In particular, within the EXPERIENCE protocol, subjects were able to complete the circuit with the exoskeleton, carrying an overall weight equal to 7 kg on their back (e.g., battery weight). The presence of the exoskeleton does not significantly alter the physiological measurements when in a seated position, while a substantial increase in the heart rate and respiration rate occurs during walking, as it appears from Figure 3. Comparing this increase with standard load-carrying activities without exoskeletons, the heart rate, heart rate variability, and respiration rate are comparable with findings in [14,15,16] and moderately higher than walking without carrying loads [17], thus suggesting room for improvement in the current version of the exoskeleton by adding assistance at the ankle joint.

Regarding psychophysiological metrics shown in Figure 4, the exoskeleton does not affect either the stress or the attention of the user, meaning that no mental workload occurs while wearing the exoskeleton. This result pushes forward the actual possibility of using robotic systems in load-carrying operative conditions. Subject 1 shows higher differences in terms of fatigue, energy, HRV, and GSR compared to other subjects. Since all metrics depend on the performed measurements, these are also strongly correlated with the emotional state of the subject, becoming very subjective and, thus, very susceptible to differences among subjects. Given the low sample size of tested subjects, these differences already reflect those of a potentially wider population and are of higher statistical interest. In the literature, the mental workload of using robots and exoskeletons has often been linked to an increase in physiological measures such as heart rate and heart rate variability, or to a decrease in galvanic skin response [18]. However, in the case of analyses in which a physical activity is included, there is no significant correlation between these measures and a potential mental workload given the overlapping behavior of these metrics in these contexts. Instead, fatigue and energy expenditure occur after reaching a steady state during walking, coherently linked to the increase in physiological measures as aforementioned. Questionnaire answers reported in Figure 5 and Section 3, although subjective, revealed positive feedback, on average, on all four PIs, especially on the acceptability and perceptibility ones. Usability was scored less, on average, given the absence of useful DoFs (e.g., hip abduction/adduction or ankle lateral/medial rotation) on the exoskeleton, which affected the gait pattern.

In conclusion, the benchmarking platform successfully assessed the performance of the *Wearable Walker* lower limb exoskeleton, revealing the efficacy of the control strategy and a low mental workload, but also showing that further improvements from the assistive and constructive design point of view should be carried out.

Future research will involve testing an actuated ankle to explore advanced control possibilities and study balance control, as described in [19]. A reduced number of actuated joints in a lower limb exoskeleton decreases the device’s cost, making it more marketable. In fact, most existing exoskeletons do not include ankle actuation. However, studies on human and humanoid posture control and balance have emphasized the significance of ankle torque for standing [20,21,22,23]. Comparing configurations with and without ankle actuation could clarify the ankle joint’s role and inspire the development of self-balancing exoskeletons or those that assist with balance. The ability to switch smoothly between two controllers is an important feature of the *Blend Control* (Section 2.2). This will benefit from future improvements in the identification of the exoskeleton state, such as those reported in [9,24,25,26].

## Figures and Tables

**Figure 1 sensors-24-05358-f001:**
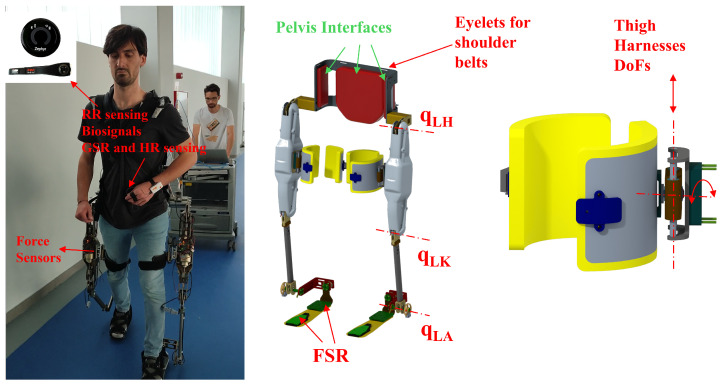
The *Wearable Walker* lower limb exoskeleton along with the sensing devices used for the experiment. The CAD models show the kinematic variables and the detail of the two degrees of freedom that the thigh harness has with regards to the exoskeleton’s thigh link.

**Figure 2 sensors-24-05358-f002:**
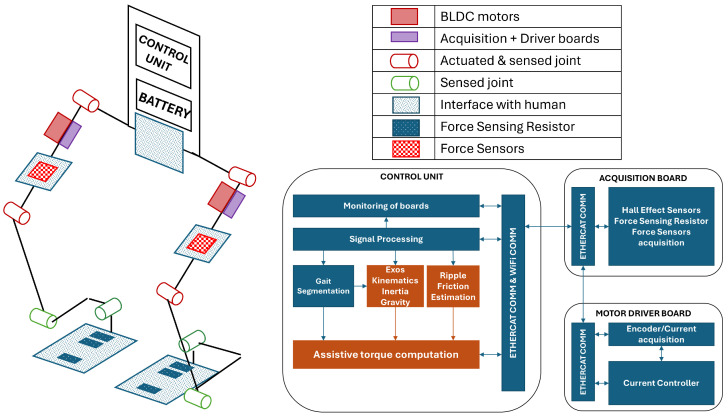
Electronics, computing units, and assistance computation architecture of the *Wearable Walker* lower limb exoskeleton. In the bottom right scheme, red blocks highlight the components of assistive torques reported in Equation (Equation 2).

**Figure 3 sensors-24-05358-f003:**
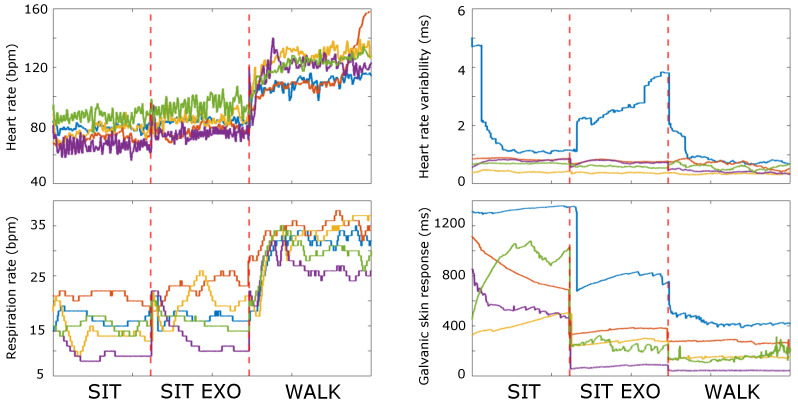
Scores of the physiological PIs of the EXPERIENCE protocol for each volunteer as a function of time. Each color corresponds to one volunteer. The three protocol phases, i.e., SIT, SIT EXO, and WALK are plotted together and marked in the figures.

**Figure 4 sensors-24-05358-f004:**
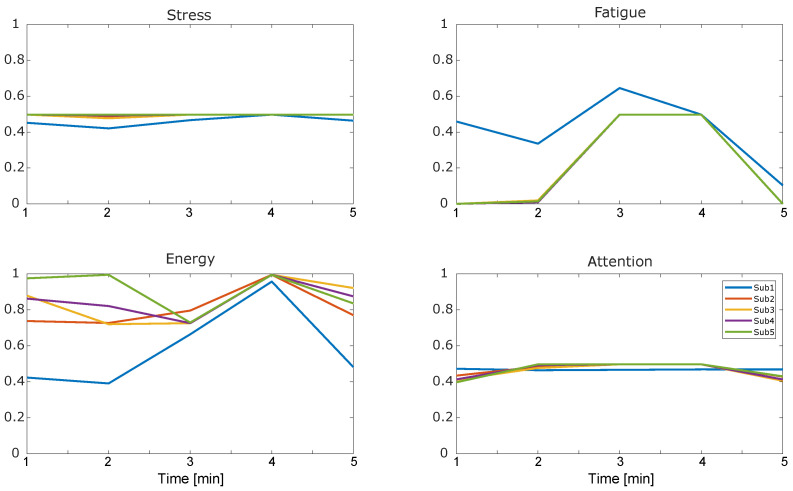
Scores for the psychophysiological performance indicators (PIs) of the EXPERIENCE protocol for each volunteer. Each point on the *x*-axis represents an average of a 1-minute recording. Curves show the evolution in time of such PIs.

**Figure 5 sensors-24-05358-f005:**
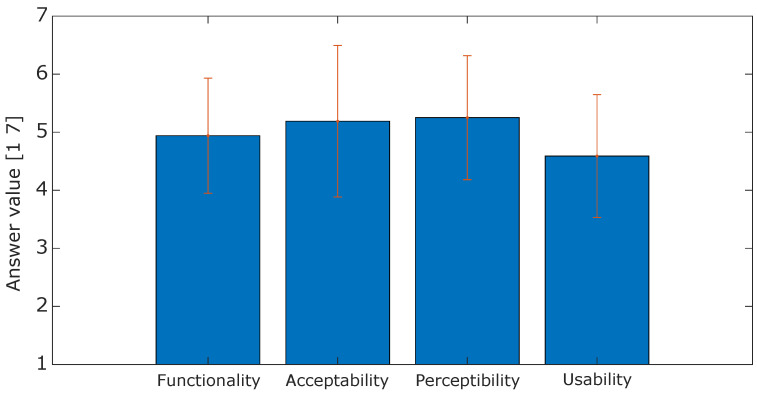
Distribution of answers to the questionnaire items. Bars report the average score and the whiskers their standard deviation.

**Table 1 sensors-24-05358-t001:** Mass and geometry information for the *Wearable Walker*. Lengths that are referred to include backlink width, thigh, and shank joint-to-joint axis distances, as well as ankle height to the foot ground contact plane.

Link	Back	Thigh	Shank	Foot
Length [m]	0.474	0.407	0.402	0.095
Mass [m]	1.2–8	4.1	2.9	0.2

## Data Availability

The data presented in this study are available on request from the corresponding author. The data are not yet publicly available at the time of writing, but they will be.

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
