# Peer review of "User-Centered Evaluation of the Wearable Walker Lower Limb Exoskeleton; Preliminary Assessment Based on the Experience Protocol"

_sensors, 2024, doi:10.3390/s24165358_

Round 1
Reviewer 1 Report
Comments and Suggestions for Authors
The application of Blend Control method in lower limb exoskeletons has shown good results in experiments. By running two modes simultaneously, with the left and right feet grounded, a smooth assist torque can be provided to prevent sudden changes. In the experiment, the performance of the exoskeleton was positively evaluated, especially in terms of acceptability and perceptibility. The analysis of physiological parameters shows that the exoskeleton does not significantly increase the pressure on the subject while sitting, but the heart rate and respiratory rate increase while walking. The analysis of psychological and physiological indicators indicates that exoskeletons do not increase subjects' stress or attention. Overall, exoskeletons have shown good performance in assisting walking, but there is still room for improvement in ankle joint drive and balance control. Future research will include testing and driving ankle joints to explore advanced control possibilities, and studying balance control to further improve the performance and subject experience of exoskeletons. But there are still some problems as follows:
1. It is recommended to unify the formulas in the article, such as removing the "{" in formula 2 or adding "{" in formulas 1 and 4;
2. It is suggested to add images to assist in the description of pages 65-68 on page 3;
3. On page 3, lines 85-86, does the efficiency of over 90% refer to the tendon-driven transmission system ? Why can good backdrivability be guaranteed;
4. It is suggested to supplement the meaning of " ˙q " in formula (2) on page 4;
5. It is recommended to adjust the x-axis. Because in the description on page 7, line 258, the four indicators of psychophysiological PI were considered for the last 15 minutes, but only five minutes were described in Figure 2;
6. Sub1 shows fatigue and energy in Figure 2; There are significant data differences between HRV and GSR in Figure 3 and the other four subjects. Please provide additional explanation.
Reviewer 2 Report
Comments and Suggestions for Authors
In this paper, a lower limb exoskeleton named Wearable Walker is described and preliminarily evaluated using Experience protocol. Wearable Walker is an innovative lower limb exoskeleton designed to help stroke patients with daily activities while enhancing human capabilities, especially in load-carrying tasks. This paper introduces the physical characteristics, control strategy and sensing ability of Wearable Walker, and describes the experimental protocol of EXPERIENCE subproject in detail. The experimental protocol involved walking on level ground and collecting physiological signals, such as heart rate, respiratory rate and electrodermal response, and completing questionnaires. The purpose of the article is to evaluate Wearable Walker's specific exoskeleton and current control systems to gain insights for improvements and to provide a case study of a replicable wearable robot benchmark. The paper also introduces the use of questionnaires to obtain four evaluation indicators of usability, acceptability, perceptibility and functionality.
The paper is logical and the experimental setup is sound, but there are still some problems. Deficiencies and possible directions for improvement of this article include:
1. In section 1, the layout of Table 1 separated the content of the last paragraph and needed to be adjusted; The reference to Table 1 in the paper is in the third paragraph of Section 2.1. Does the position of Table 1 need to be adjusted?
2. In section 2.1, it is suggested to include a flowchart of the operation of the wearable Walker exoskeleton for easy reader understanding.
3. In section 2.3.2, we describe the advantages of testing with ZephyrBioModule 3 sensors and Shimmer sensors over other sensor tests to improve the accuracy of test results.
4. Are the positions placed in sections 2.3.3 and 2.3.4 correct? The overall layout of section 2.3 is inconsistent.
5. Both figures 2 and 4 have the problem of separating entire paragraphs in the text.
6. In the conclusion, it is suggested to use the specific data of the experiment to illustrate the results of the evaluation.
7. Whether the captions of the drawings in the paper should be centered.
